# Auxin/Cytokinin Antagonistic Control of the Shoot/Root Growth Ratio and Its Relevance for Adaptation to Drought and Nutrient Deficiency Stresses

**DOI:** 10.3390/ijms23041933

**Published:** 2022-02-09

**Authors:** Jasmina Kurepa, Jan A. Smalle

**Affiliations:** Plant Physiology, Biochemistry, Molecular Biology Program, Department of Plant and Soil Sciences, University of Kentucky, Lexington, KY 40546, USA; jasmina.kurepa@uky.edu

**Keywords:** auxin signaling, cytokinin signaling, auxin/cytokinin signaling crosstalk, hormone antagonism, drought stress, nutrient deficiency, shoot/root growth ratio

## Abstract

The hormones auxin and cytokinin regulate numerous aspects of plant development and often act as an antagonistic hormone pair. One of the more striking examples of the auxin/cytokinin antagonism involves regulation of the shoot/root growth ratio in which cytokinin promotes shoot and inhibits root growth, whereas auxin does the opposite. Control of the shoot/root growth ratio is essential for the survival of terrestrial plants because it allows growth adaptations to water and mineral nutrient availability in the soil. Because a decrease in shoot growth combined with an increase in root growth leads to survival under drought stress and nutrient limiting conditions, it was not surprising to find that auxin promotes, while cytokinin reduces, drought stress tolerance and nutrient uptake. Recent data show that drought stress and nutrient availability also alter the cytokinin and auxin signaling and biosynthesis pathways and that this stress-induced regulation affects cytokinin and auxin in the opposite manner. These antagonistic effects of cytokinin and auxin suggested that each hormone directly and negatively regulates biosynthesis or signaling of the other. However, a growing body of evidence supports unidirectional regulation, with auxin emerging as the primary regulatory component. This master regulatory role of auxin may not come as a surprise when viewed from an evolutionary perspective.

## 1. Introduction

Adaptive regulation of the shoot/root growth ratio is an evolutionarily conserved developmental mechanism in terrestrial plants, which ensures maximal progeny production under fluctuating environmental conditions. In growth conditions in which water and mineral nutrients are not limiting, shoot growth is favored as it embodies traits that support reproductive success and survival of the species, and root development is limited to a level sufficient to sustain shoot development without the unnecessary depletion of photosynthates [1]. Thus, under optimal water and nutrient conditions, plants are predicted to have a high shoot/root ratio [1,2]. However, when water or mineral nutrient availability decreases, the growth of the shoot—the main site for water loss and mineral nutrients consumption—needs to be reduced in favor of the growth of a larger root system [3,4,5].

This review outlines the antagonistic roles played by cytokinin and auxin in controlling the shoot/root growth ratio and focuses on the link between this antagonism and the adaption to drought stress and nutrient deficiency, two of the main environmental challenges for terrestrial plants.

## 2. Auxin- and Cytokinin-Dependent Control of the Shoot/Root Growth Ratio

### 2.1. Auxin and Cytokinin Signaling Pathways

Auxin signaling is essentially an inhibition–release mechanism (Figure 1). The main components of this signaling pathway are the auxin resistant/indole-3-acetic acid inducible (AUX/IAA) proteins, auxin response factor (ARF) proteins, and F-box proteins of the transport inhibitor response 1/auxin signaling F-box (TIR1/AFB) family [6]. F-box proteins are substrate recruiting modules of SCF (SKP1-Cullin 1-F-box protein) E3 ligase complexes and they initiate the degradation of the substrate by the ubiquitin/26S proteasome pathway [7]. In the auxin signaling pathway, auxin acts as a ligand that promotes binding of the AUX/IAA proteins with SCF^TIR1/AFB^ which leads to the degradation of AUX/IAAs [6]. As a result, ARFs, which are inhibited by the AUX/IAAs, become activated and regulate the expression of primary auxin response genes [8]. The *AUX/IAAs* genes themselves are auxin-induced [9] and thus serve as negative feedback regulators of the auxin response. 

The cytokinin signaling pathway starts with the binding of cytokinin to histidine kinase receptors (CHKs, [10,11,12]), which leads to receptor autophosphorylation (Figure 1). The phosphoryl group is then transferred to histidine phosphotransfer proteins (HPTs), which further relay the phosphoryl group to two groups of response regulators (RRs), the type-B and type-A RRs. The phosphorylated type-B RRs are transcription factors that regulate the expression of primary cytokinin response genes that include genes encoding the type-A RRs, which function as cytokinin response inhibitors and, thus, as negative feedback regulators of the cytokinin response [13,14,15,16,17,18]. 

### 2.2. Auxin and Cytokinin Control of the Shoot/Root Growth Ratio in Higher Plants

Pioneering tissue culture experiments have shown that callus can be generated from explants using combined treatment with auxin and cytokinin, and that increasing the auxin to cytokinin ratio promotes root development, whereas decreasing the auxin to cytokinin ratio leads to shoot formation [19]. These experiments, now a cornerstone of basic and applied plant research, were the first to show that auxin is a root growth-promoting hormone, whereas cytokinin is a shoot growth promoter [19]. 

Subsequent work with mutant and transgenic lines provided unequivocal evidence that auxin and cytokinin have opposite effects on root and shoot growth. Both root and shoot growth are inhibited in Arabidopsis transgenic lines with a strong constitutive auxin or cytokinin response [20,21]. Strong auxin or cytokinin resistance is also associated with severe root and shoot growth inhibition [12,22,23]. However, studies with mutant and transgenic lines that have weaker changes in hormone action, proved that auxin and cytokinin antagonistically regulate the shoot/root growth ratio. For example, transgenic plants with lower cytokinin content have a smaller shoot and a larger root system than the wild type [24,25]. The root phenotype of transgenic plants with lower cytokinin content could be viewed as compensatory (more nutrients may be available for root growth if shoot growth is reduced). This compensatory hypothesis was refuted by analyses of Arabidopsis and tobacco plants in which cytokinin content was reduced only in roots [26]. These transgenic plants have an enlarged root system and a normal wild type-sized shoot, proving that cytokinin directly represses root growth [26]. A shoot-growth promoting effect of cytokinin was also described for a range of grass species that have increased cytokinin content due to elevated expression of *STENOFOLIA (STF),* which encodes a member of the WUSCHEL-related homeobox (WOX) family of transcription factors that represses the expression of cytokinin oxidase/dehydrogenase genes [27]. Moreover, a study with the aquatic plants *Lemna gibba* and *Spirodela polyrhiza* revealed that cytokinin treatment promoted frond (i.e., shoot) expansion and duplication while suppressing root elongation [28]. In addition, research on potato stem single-node cuttings revealed that whereas cytokinin treatments increased, auxin treatments decreased the shoot/root growth ratio [29]. These conclusions reached by analyses of cytokinin-treated plants and transgenic lines with altered cytokinin content were confirmed by cytokinin receptor mutant studies. The Arabidopsis histidine kinase receptor (AHK) double mutant *ahk2 ahk3* has smaller shoots and a larger root system [30], and gain-of-function AHK2 or AHK3 plants have enlarged shoots and smaller root systems when compared to wild-type plants [31]. 

In contrast to cytokinin resistant mutants and transgenic lines with lower cytokinin content, Arabidopsis auxin resistant (*axr*) mutants have an increased shoot/root biomass ratio (Figure 2). The *axr3*, *axr2*, and *axr5* mutants, which have decreased auxin sensitivity because of the stabilization of auxin response repressor AUX/IAA proteins [32,33,34], have been ranked based on the strength of their auxin resistance [35]. The strongest *axr* mutant, *axr3-3*, has a nearly five-fold increase in shoot/root biomass ratio (Figure 2). However, it could be argued that the overall growth retardation of *axr3-3* plants precludes any relevant comparison with the wild type. However, the shoot/root biomass ratio was also increased in the medium-strength auxin insensitive mutant *axr2* and the weakest mutant *axr5,* indicating that a mild decrease in auxin action increases the shoot/root growth ratio without causing the overall dwarfism that characterizes strong auxin resistant mutants such as *axr3-3* (Figure 2). Moreover, mutant plants defective in auxin biosynthesis also have an increased shoot/root growth ratio, which confirms that auxin and cytokinin antagonistically regulate the growth rate of aerial and root organs [36]. 

### 2.3. Auxin- and Cytokinin-Dependent Control of the Growth Ratio of Shoot/Root Equivalents in Bryophytes

Auxin and cytokinin play key roles in vascular system development and in part, exert their effects on plant growth via vascular transport [37,38,39,40,41,42,43]. However, the antagonistic actions of auxin and cytokinin on the shoot/root growth ratio are operational in the earliest land plants, the non-vascular bryophytes, which suggests that it predates the emergence of the vascular system. Key to the evolution of land plants was the establishment of a program that controls the development of an upward-growing and photosynthesizing shoot-like organ on the one hand, and downward-growing root-like cells on the other [44,45]. Bryophytes have no shoots and roots but have developmentally equivalent organs. In the liverwort *Marchantia polymorpha*, the thallus is the photosynthesizing part, and it forms gemma cups, multicellular asexual buds, that can be considered the functional equivalents of lateral shoots in higher plants [46]. The flat thallus has dorsal and ventral parts, with the dorsal part responsible for gemma cup formation, while the ventral part produces rhizoids that anchor the liverwort to the soil and take up nutrients and water [47]. Thus, in liverworts, the dorsal part of the thallus and its upward growing structures are the shoot equivalent, and the ventral part of the thallus and the rhizoids are equivalent to roots. Accordingly, an increase in the “shoot/root” growth ratio in *M. polymorpha* is represented by an increase in the growth rate of the dorsal part of the thallus, combined with an increase in gemma cup initiation and growth, which is a growth pattern that leads to the development of an epinastic (i.e., downward-bending) thallus. A decrease in the “shoot/root” growth ratio, resulting from increased growth of the ventral thallus combined with increased rhizoid formation and growth, will cause hyponastic (i.e., upward-bending) thallus growth. 

Functional auxin and cytokinin signaling and biosynthesis exist in both the liverwort *M. polymorpha* and the moss *Physcomitrium patens* (previously known as *Physcomitrella patens*) [48,49,50,51,52,53]. The signaling and biosynthetic pathways in *M. polymorpha* and *P. patens* have the same core components as Arabidopsis, albeit encoded by less complex gene families [48,49,50,51,52,53]. Strikingly, hormone treatment studies and analyses of *M. polymorpha* mutant and transgenic lines with altered hormone content or sensitivity, revealed that auxin and cytokinin affect the growth ratio of the shoot and root equivalents in the same manner as in higher plants: increased cytokinin action promotes gemma cup initiation and causes epinastic thallus growth [46,49,54,55], and increased auxin action leads to the formation of a hyponastic thallus with an increased number of larger rhizoids and a decreased gemma cup initiation rate [46,50,51,56]. In the moss *P. patens*, the gametophores and rhizoids that develop from the caulonema stage can be considered the equivalents of shoots and roots in higher plants. The gametophore contains photosynthesizing leaves and represents the *P. patens* reproductive phase, whereas the rhizoids anchor the moss to the soil and take up nutrients and water [53,57]. Similar to higher plants and the liverworts, auxin and cytokinin exert opposite effects on the development of gametophores and rhizoids: auxin treatment suppresses gametophore development and causes the formation of ectopic rhizoids, whereas exogenous cytokinin or a mutational increase in cytokinin content increase gametophore development and inhibit rhizoid growth [58,59,60,61,62,63,64]. 

## 3. Auxin and Cytokinin Control of the Responses to Water and Nutrient Availability 

The transition from an aquatic to a terrestrial habitat required the evolution of adaptive mechanisms needed to overcome drought stress and mineral nutrient deficiency, two main challenges associated with survival on land [65,66]. Auxin and cytokinin are essential regulators of these adaptive mechanisms, not only because they mold the shoot/root growth to a pattern that is favorable for a terrestrial lifestyle, but also because they regulate physiological processes needed to alleviate and withstand drought stress and mineral nutrient deficiency (Figure 3). 

### 3.1. Drought Stress 

Drought is one of the most important environmental stresses and together with oxidative stress, one of the most widely studied stress conditions in plant biology [67]. Drought stress is often described as multidimensional, because it causes extensive changes in plant morphology, physiology, and biochemistry [67]. Many of these changes are regulated by cytokinin and auxin. The most essential cytokinin- and auxin-regulated morphological change is a decrease in the shoot/root growth ratio, which effectively reduces the part of the plant responsible for water loss and increases the part of the plant responsible for water uptake. The most important physiological change is the antagonistic auxin- and cytokinin-dependent control of abscisic acid (ABA) biosynthesis and ABA responses. ABA, a hormone that regulates specific aspects of plant development (e.g., seed development and germination), is required for drought stress responses and drought tolerance [68,69].

#### 3.1.1. Auxin Promotes Drought Stress Tolerance

Although auxin remains best known as a growth regulatory hormone, its functions in regulating drought stress responses are now well established [70]. Drought stress induces auxin biosynthesis and alters conjugation, catabolism, transport, and response regulation, with an overall effect of increasing auxin action [71,72,73,74,75,76,77,78,79]. In addition, auxin treatments or transgenic increases in auxin content, promote drought stress tolerance in a range of plant species [71,72,73,74,75,76,77,78,79], and loss of function of the auxin biosynthesis pathway causes drought stress hypersensitivity [36,74]. Curiously, even some auxin-synthesizing and auxin-secreting *Pseudomonas* and *Rhizobium* strains increase plant tolerance to osmotic stress, a type of stress that also reduces water availability [79,80]. Reciprocal studies, including analyses of drought stress responses in auxin gain- and loss-of-function mutants and transgenic lines, confirmed that auxin positively regulates drought stress tolerance [36,74].

The auxin response pathway is required for auxin-dependent promotion of drought stress tolerance, which suggests that auxin acts as a stress response signaling molecule [81,82,83]. Although a wide range of auxin-dependent responses to drought stress has been described, all the responses ultimately impact water uptake and strengthen the protection against dehydration damage. For example, a decrease in soil water content alters root growth in an auxin-dependent manner, by increasing the initiation and growth of lateral roots and promoting elongation of the primary root [82]. The increased lateral-root-growth response to drought stress was shown to depend on auxin regulation at the biosynthesis, transport, and signaling levels [82]. Moreover, osmotic stress was shown to inhibit leaf expansion growth by increased auxin action via the ARF family of auxin response activators, indicating that auxin also mediates drought stress responses in the shoot [84]. Auxin may also positively impact drought tolerance by limiting stomatal density and controlling stomatal aperture [85]. 

Current research points to the auxin-inducible *AUX/IAA* genes as important regulators of auxin-induced drought stress tolerance. Overexpression of AUX/IAA genes was shown to promote drought tolerance in rice and tobacco plants [86,87], and in rice, this was accompanied by increased expression of members of the YUCCA family of auxin biosynthesis genes [86]. AUX/IAAs were also found to promote drought stress tolerance by the regulation of glucosinate and the promotion of stomatal closure [88]. AUX/IAAs are generally auxin response inhibitors [89,90], but their positive role in promoting auxin-mediated drought stress tolerance suggests that they have a more complex role in auxin signaling. Another family of auxin-inducible genes, the *Small Auxin-Up RNAs (SAURs),* was also implicated in promoting drought (and salt) stress tolerance [91]. *SAUR* genes are ABA-inducible, and they mediate, at least in part, ABA-induced stomatal closure and seed germination [91]. Thus, the *SAUR* gene family is viewed as an auxin interaction point with ABA signaling.

#### 3.1.2. Cytokinin Negatively Regulates Drought Stress Tolerance

The link between cytokinin and drought stress tolerance was uncovered over the past two decades [70]. Drought stress represses cytokinin biosynthesis and down-regulates the expression of genes encoding cytokinin signaling components [92,93,94]. Further studies using mutant and transgenic lines revealed that loss of function of components of the cytokinin response pathway, cytokinin resistance or decreased cytokinin content are associated with increased drought stress tolerance, whereas increased cytokinin action causes drought stress hypersensitivity [92,93,95,96,97,98,99], which confirmed that cytokinin negatively affects drought stress tolerance. 

A plethora of cytokinin responses that lead to a decrease in drought stress tolerance has been described. Most notably, cytokinin increases water loss by promoting shoot growth and limits water uptake by inhibiting root growth [25,26]. In addition, cytokinin suppresses drought stress tolerance by repressing the expression of ABA-inducible genes [97]. On the other hand, decreased cytokinin content was associated with increased membrane integrity under drought stress conditions and increased ABA sensitivity [93]. While cytokinin treatments have been shown to increase stomatal density in leaves, cytokinin is also known to counteract the effect of ABA on stomatal closure [85,100]. Recently, it was discovered that this cytokinin/ABA crosstalk regulates osmotic stress tolerance by regulating the global control of protein synthesis rates [101]. Transgenic lines that have increased cytokinin action due to increased activity of the Arabidopsis Type-B response Regulator 1 (ARR1) are hypersensitive to osmotic stress [101]. This hypersensitivity is the result of an increase in global protein synthesis, which is—at least partly—caused by the increased expression of cytokinin-inducible *RPL4A* and *RPL4D* genes that encode ribosomal protein L4 (RPL4) isoforms A and D [101]. ABA, a known repressor of protein synthesis [102], successfully suppressed the osmotic stress hypersensitivity of ARR1 gain-of-function lines [101]. 

The salt stress and drought stress responses are closely related, which is not surprising considering that salt stress leads to a reduction in water availability [103,104]. The mitogen-activated protein kinases 3 and 6 (MPK3 and MPK6) play an essential role in the salt stress response of plants, and they promote salt stress tolerance by promoting degradation of Type-B RR cytokinin response activators [105]. It is still to be determined if MPK3- and MPK6-dependent degradation of Type-B RRs plays a role in drought stress response, but if it does, it will bring to light another mechanism of drought stress-dependent suppression of cytokinin signaling. 

In summary, the negative role of cytokinin in drought stress tolerance is now well established and supported by cytokinin mutant studies and analyses of transgenic plants with decreased cytokinin content or increased action [92,93,96]. Furthermore, as decreased cytokinin content was also shown to promote drought stress tolerance in the moss *P. patens*, this negative effect of cytokinin on drought stress tolerance may be conserved throughout the plant kingdom [106]. However, cytokinin treatments or transgenic increases in cytokinin content have also been shown to promote drought stress tolerance [107,108,109,110]. One possible explanation for these contradictory results involves the role played by cytokinin in protecting the photosynthetic machinery under stress conditions [110,111,112]. Cytokinin promotion of drought stress tolerance has been successful when cytokinin biosynthesis was increased in response to drought stress and in senescing leaves of plants expressing a cytokinin biosynthesis transgene from a senescence- or stress-inducible promoter [110,111,112,113,114,115,116,117]. These approaches circumvent the negative drought stress impact of cytokinin on developing leaves, wherein cytokinin promotes water loss by increasing stomatal density and aperture [85,100]. Therefore, the timing of cytokinin treatment seems to be crucial for promoting this type of drought stress tolerance.

### 3.2. Mineral Nutrient Availability

Since agricultural productivity is inseparably linked to nutrient availability, research on nutrient uptake and responses to nutrient deficiency has been extensive and encompassed a myriad of research directions from soil chemistry to molecular studies of different transporter proteins. The two main limiting nutrients in soils, nitrogen and phosphate, are of particular interest here, as the antagonistic effects of cytokinin and auxin on their uptake and the responses to their decreased availability has been well-documented. 

#### 3.2.1. Auxin Positively Regulates Nutrient Uptake

The nitrogen content of soils impacts auxin accumulation and auxin-dependent root growth. Whereas severe nitrogen deficiency inhibits the growth of both the primary and lateral roots [82], mild nitrogen deficiency increases auxin content and signaling, leading to an increase in lateral root formation, lateral root growth, and elongation of the primary root [118,119]. In Arabidopsis, this auxin-dependent promotion of lateral root formation acts in an ion-dependent manner, with ammonium promoting lateral root branching and nitrate promoting lateral root elongation [120,121,122,123,124,125]. 

Similar to nitrogen, phosphate deficiency triggers lateral root formation by increasing auxin sensitivity and biosynthesis [126,127]. On the other hand, auxin induces the expression of Phosphate Starvation Response 1 (PHR1), a key activator of phosphate starvation response genes, such as genes that encode phosphate transporters [126,127,128,129,130]. 

#### 3.2.2. Cytokinin Is a Satiation Hormone

Cytokinin generally acts as a negative regulator of mineral nutrient uptake [131,132,133,134]. Nitrogen deficiency was shown to decrease cytokinin content by simultaneously inhibiting cytokinin biosynthesis and promoting cytokinin degradation, thus increasing root growth and mineral nutrient accumulation [135]. The opposite was also shown to hold true: increased nitrogen availability promotes cytokinin accumulation and a concomitant reduction in root growth [136,137,138]. Furthermore, the finding that cytokinin treatments repress genes involved in nitrogen uptake adds to the understanding of the molecular mechanisms by which cytokinin negatively regulates nitrogen uptake [131]. Cytokinin also suppresses phosphate uptake by repressing the genes involved in the phosphate starvation response, and this negative gene regulation requires the cytokinin response pathway [139,140]. 

Although cytokinin negatively regulates mineral nutrient uptake, it is important to note that this regulation is only active in response to nutrient excess. As previously stated, nutrient deficiency negatively regulates cytokinin action. Essentially, cytokinin serves as a satiation hormone as it prevents nutrient accumulation to excessive and potentially toxic levels [131].

## 4. Unidirectional Control of Auxin/Cytokinin Antagonism 

The antagonistic auxin/cytokinin regulation of fundamental processes such as the shoot/root growth ratio and water and nutrient uptake implies the existence of a checkpoint(s) or negative feedback loop(s) by which one hormone regulates the biosynthesis, signaling, or transport pathway of the other (Figure 4). Auxin/cytokinin interactions have been investigated in detail in both shoot and root apical meristems, where they play a key role in the initiation of leaves and the priming of lateral roots and thus, determine shoot and root architecture [141,142,143,144,145,146]. Inhibitory auxin/cytokinin interactions have also been described for other phases of plant development (e.g., leaf development, shoot and root elongation, and the branching of shoots and roots). These interactions control the shoot to root growth ratio, and they are predominantly unidirectional, with auxin inhibiting cytokinin, but not the other way around. Auxin-dependent negative control has been described for cytokinin biosynthesis and signaling.

### 4.1. Auxin Inhibition of Cytokinin Biosynthesis

The inhibitory control of auxin on cytokinin biosynthesis involves the canonical auxin response pathway and is functional in whole plants and in specific organs (e.g., apical meristems and stem internodes) [147,148,149,150,151]. In whole plants, auxin repression predominantly involves inhibition of the isopentenyladenosine-5′-monophosphate-independent pathway of cytokinin biosynthesis [150]. In contrast, cytokinin promotes auxin biosynthesis throughout the plant [152,153,154,155]. For example, high-level expression of a gene encoding the cytokinin biosynthetic enzyme isopentenyltransferase 8 was shown to increase auxin accumulation in the shoot apex and in developing leaves and roots. In contrast, the downregulation of cytokinin content, resulting from either overexpression of a cytokinin oxidase gene or by the loss of function of cytokinin biosynthesis genes, decreased auxin content [152]. Cytokinin-induced auxin synthesis is—at least in part—promoted by cytokinin-dependent upregulation of genes involved in auxin biosynthesis, and it requires the cytokinin response pathway [152,153,154,155]. Considering that auxin is a cytokinin biosynthesis repressor, it is tempting to speculate that cytokinin-induced auxin synthesis serves as a feedback control mechanism that limits cytokinin accumulation and thus cytokinin action. 

### 4.2. Auxin Inhibition of Cytokinin Signaling and Action

It was recently shown that whereas auxin inhibits cytokinin signaling in both shoot and root organs, cytokinin does not negatively affect auxin signaling [35]. Auxin resistance caused by the stabilization of AUX/IAA proteins was associated with increased cytokinin signaling in a mutant strength-dependent manner [35]. Moreover, loss-of-function of the auxin response activator gene *ARF7* also combined auxin resistance with increased cytokinin action [35]. In addition to a constitutive increase in cytokinin signaling, auxin resistance was associated with a hypersensitive response to exogenous cytokinin, proving that auxin impacts cytokinin signaling along with cytokinin biosynthesis [35]. In contrast, severe cytokinin resistance caused by the simultaneous loss of function of the ARR1, ARR10, and ARR12 cytokinin response activators did not impact auxin signaling intensity [35], indicating that this antagonistic auxin/cytokinin interaction at the signaling level is also unidirectional. Two recent reports have provided further evidence for this conclusion [147,156]. These studies show that transgenic increase in endogenous auxin and auxin treatments of potato plants, decrease cytokinin content and strongly repress cytokinin signaling genes, whereas cytokinin treatment leads to an ambiguous and sucrose-dependent effect on auxin signaling [147,156].

## 5. Evolutionary Implications and Future Perspectives 

When viewed from an evolutionary perspective, the auxin/cytokinin antagonism and the unidirectional control of cytokinin by auxin allow land plants to flexibly control their shoot/root growth ratio, which is essential for survival in a terrestrial habitat. As auxin and cytokinin also exert antagonistic effects on the growth ratio of shoot and root equivalents in the bryophytes, it will be interesting to determine if the unidirectional auxin inhibition of cytokinin biosynthesis and signaling is also functional in these earliest land plants. In order to gain a better understanding of the mechanisms driving the evolution of land plants, it will also be important to test whether the auxin/cytokinin antagonism controls drought stress and nutrient deficiency responses in bryophytes. If this is the case, it would be of interest to determine whether cytokinin biosynthesis and signaling developed independently from auxin control and were connected to and restricted by auxin during the evolution of land plants, or if they developed co-dependently, for example, as modules of the original auxin response. The Charophyte green algae share their ancestral lineage with land plants [157]. In some Charophytes, auxin and cytokinin have been detected, and orthologues of auxin and cytokinin biosynthesis and response pathway genes have been identified [158,159,160]. It will be of interest to determine whether auxin and cytokinin pathways are linked or function independently in these land plant progenitors. 

## Figures and Tables

**Figure 1 ijms-23-01933-f001:**
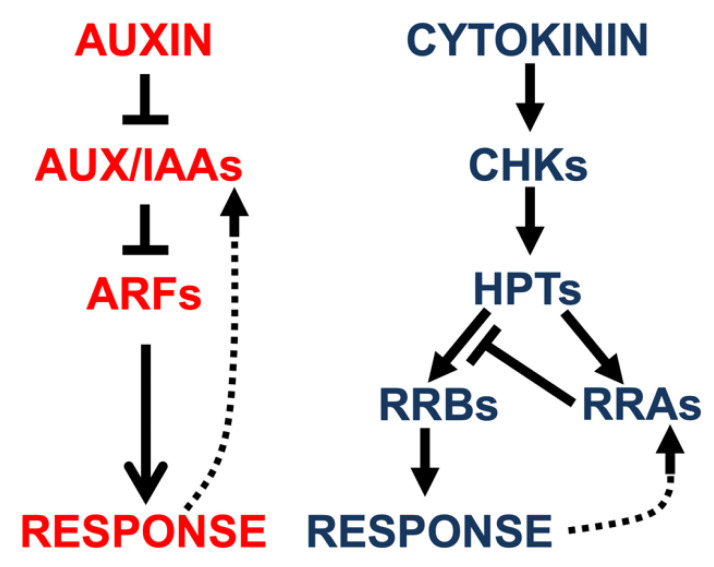
Simplified schemes of the auxin and cytokinin response pathways. Abbreviations: AUX/IAAs, auxin/indole-3-acetic acid regulators; ARFs, auxin response factors; CHKs, cytokinin histidine kinase receptors; HPTs, histidine phosphotransfer proteins; RRBs, type-B response regulators; RRAs, type-A response regulators.

**Figure 2 ijms-23-01933-f002:**
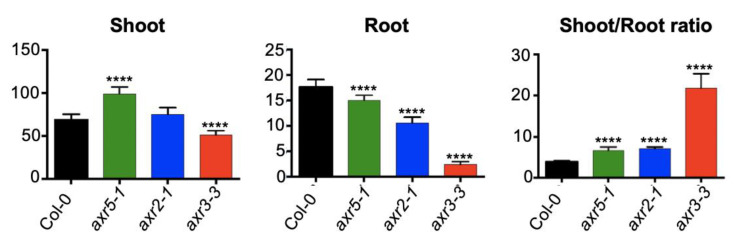
Increased shoot/root growth ratio in Arabidopsis auxin resistant mutants. Nineteen-day-old Col-0 wild type, *axr5-1*, *axr2-1*, and *axr3-3* plants, grown on vertical plates containing half-strength Murashige and Skoog medium, were dissected, and the fresh weights of shoots and roots were measured. Data are presented as mean ± SD (n ≥ 12 pools of 8 plants each). The significance of the difference between the wild type and the mutants is noted (****, *p* < 0.0001; two-way ANOVA with Bonferroni’s multiple comparisons test) *Kurepa et al., unpublished*.

**Figure 3 ijms-23-01933-f003:**
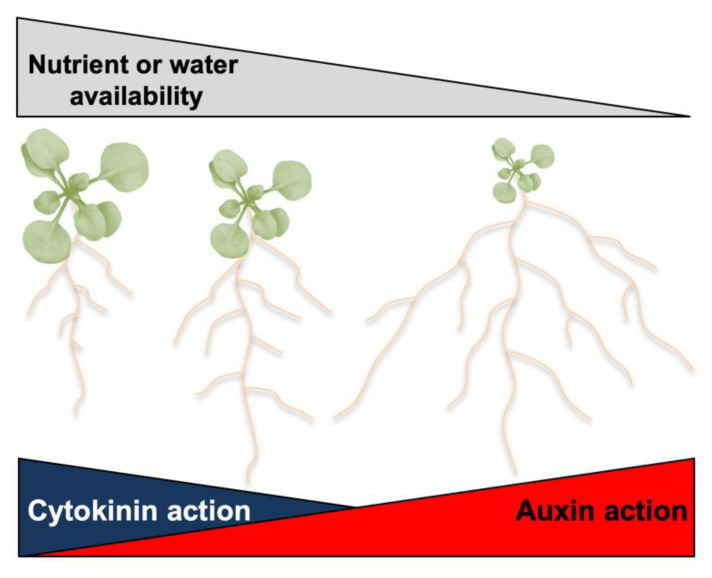
Effects of nutrient or water availability and effects of the auxin/cytokinin ratio on the shoot/root growth ratio.

**Figure 4 ijms-23-01933-f004:**
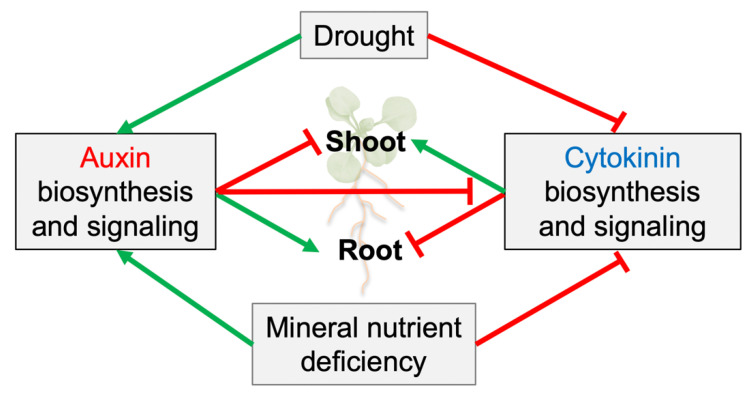
Schematic summary of the antagonistic cytokinin/auxin control of the shoot/root ratio in relation to water deficit and nutrient deficiency.

## Data Availability

The data is contained within the article.

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
