# Peer review of "Auxin/Cytokinin Antagonistic Control of the Shoot/Root Growth Ratio and Its Relevance for Adaptation to Drought and Nutrient Deficiency Stresses"

_ijms, 2022, doi:10.3390/ijms23041933_

Round 1
Reviewer 1 Report
It is a good review on the topic.
Comments and corrections that must be made:
Figure 1. Since in paragraph 2.1 (lines 55-70) auxin signaling is first in relation to cytokinin signaling, it seems reasonable that in Figure 1 the auxin signaling scheme should be placed to the left and that of cytokinin on the right.
In general, I find errors in the numbering of paragraphs starting from line 121. For example, 2.2 is already used in line 72. Authors should carefully review this aspect in the rest of the manuscript.
Author Response
We have copied the reviewer comments and have inserted our response in italic.
Reviewer #1
It is a good review on the topic.
Comments and corrections that must be made:
- Figure 1. Since in paragraph 2.1 (lines 55-70) auxin signaling is first in relation to cytokinin signaling, it seems reasonable that in Figure 1 the auxin signaling scheme should be placed to the left and that of cytokinin on the right.
This change indeed adds to the flow of the manuscript. We have also adjusted the legend to reflect the change in the scheme.
In addition, we have replaced the images in all figures with higher-resolution images.
- In general, I find errors in the numbering of paragraphs starting from line 121. For example, 2.2 is already used in line 72. Authors should carefully review this aspect in the rest of the manuscript.
Thank you. We found a mistake in the template heading use and corrected it.
Reviewer 2 Report
The review article entitled "Auxin/cytokinin antagonisms and the shoot/root growth ratio" discussed the Auxin/cytokinin and shoot/root growth and how drought stress and nutrient availability alter the cytokinin and auxin signaling and biosynthesis pathways. The review is of relevance and general interest to the journal's readers. However, I have several concerns about presenting the data that should be addressed before publication.
- The authors are highly recommended to avoid using a personal pronoun (e.g., We, our, etc.); they can use the third party in the past tense's passive voice.
- The authors need to carefully read through the manuscript to correct typos and grammar to improve the manuscript.
- Any abbreviation must be associated with the full name at the first mention in the manuscript to allow the reader to follow up because not all the readers are familiar with the abbreviated terminology.
- The numbering of the subtitles in the manuscript is misleading
- The authors need to provide some details in section 2.1 instead of just listing the citation for that.
- In figure 2, the authors need to provide the source of these data, even if it is their own data.
Author Response
We have copied the reviewer comments and inserted our responses in italic.
Reviewer #2
The review article entitled "Auxin/cytokinin antagonisms and the shoot/root growth ratio" discussed the Auxin/cytokinin and shoot/root growth and how drought stress and nutrient availability alter the cytokinin and auxin signaling and biosynthesis pathways.
The review is of relevance and general interest to the journal's readers.
However, I have several concerns about presenting the data that should be addressed before publication.
- The authors are highly recommended to avoid using a personal pronoun (e.g., We, our, etc.); they can use the third party in the past tense's passive voice.
We have deleted the part of Section 2.1 were most of the personal pronouns were used (please see our answer to comment #5). We have also corrected the same issue in Section 3.2.
- The authors need to carefully read through the manuscript to correct typos and grammar to improve the manuscript.
We have made many a number of corrections throughout the manuscript to improve the quality.
- Any abbreviation must be associated with the full name at the first mention in the manuscript to allow the reader to follow up because not all the readers are familiar with the abbreviated terminology.
Abbreviations use is corrected.
- The numbering of the subtitles in the manuscript is misleading
We have made a mistake in template use and corrected it in this new version of the manuscript.
- The authors need to provide some details in section 2.1 instead of just listing the citation for that.
We agree that this part is somewhat confusing. We have deleted the introductory part of section 2.1. The remaining part describes the core auxin and cytokinin signaling mechanisms which is sufficient for understanding the remainder of this review.
- In figure 2, the authors need to provide the source of these data, cyt,even if it is their own data.
Corrected. Thank you.
Round 2
Reviewer 2 Report
The authors responded to all my comments!
Author Response
Thank you